# A mutation uncouples the tubulin conformational and GTPase cycles, revealing allosteric control of microtubule dynamics

Elisabeth A Geyer[1], Alexander Burns[1], Beth A Lalonde[2], Xuecheng Ye[1], Felipe-Andres Piedra[1], Tim C Huffaker[2], Luke M Rice[1]*

[1]Departments of Biophysics and Biochemistry, University of Texas Southwestern Medical Center, Dallas, United States; [2]Department of Molecular Biology and Genetics, Cornell University, Ithaca, United States

**Abstract** Microtubule dynamic instability depends on the GTPase activity of the polymerizing $\alpha\beta$-tubulin subunits, which cycle through at least three distinct conformations as they move into and out of microtubules. How this conformational cycle contributes to microtubule growing, shrinking, and switching remains unknown. Here, we report that a buried mutation in $\alpha\beta$-tubulin yields microtubules with dramatically reduced shrinking rate and catastrophe frequency. The mutation causes these effects by suppressing a conformational change that normally occurs in response to GTP hydrolysis in the lattice, without detectably changing the conformation of unpolymerized $\alpha\beta$-tubulin. Thus, the mutation weakens the coupling between the conformational and GTPase cycles of $\alpha\beta$-tubulin. By showing that the mutation predominantly affects post-GTPase conformational and dynamic properties of microtubules, our data reveal that the strength of the allosteric response to GDP in the lattice dictates the frequency of catastrophe and the severity of rapid shrinking.

*For correspondence: Luke.Rice@UTSouthwestern.edu

**Competing interests:** The authors declare that no competing interests exist.

## Introduction

Microtubules are dynamic polymers of αβ-tubulin that undergo GTPase-dependent switching between phases of growing and rapid shrinking (reviewed in *Desai and Mitchison, 1997*). Microtubule growing and shrinking rates, and switching frequencies, are necessary for proper function, highly regulated by cellular factors (*Howard and Hyman, 2003*), and targeted by widely used anti-mitotic anticancer drugs (*Jordan and Wilson, 2004*; *Amos, 2011*). αβ-tubulin also undergoes a conformational cycle during polymerization and depolymerization (*Mandelkow et al., 1991a*; *Chrétien et al., 1995*) Unpolymerized αβ-tubulin adopts a 'curved' conformation (*Ravelli et al., 2004*; *Rice et al., 2008*; *Buey et al., 2006*) that is not compatible with the 'straight' microtubule lattice (*Nogales et al., 1998*). In the body of the microtubule, αβ-tubulin can adopt distinct 'expanded' or 'compacted' straight conformations depending on nucleotide state (*Alushin et al., 2014*; *Hyman et al., 1995*). Partially straightened conformations of αβ-tubulin are thought to occur as polymerization intermediates on the microtubule end.

Recent studies have begun to reveal how microtubule regulatory proteins take advantage of these different conformations. In one well-characterized example, end-binding proteins in the EB1 family have been shown to mark the plus-end cap of microtubules by binding preferentially to the 'expanded', GTP-like conformation of αβ-tubulin that only occurs near the end of the growing microtubule lattice (*Maurer et al., 2011*; *2012*; *Zanic et al., 2009*; *Alushin et al., 2014*). Another recent

**eLife digest** Protein filaments called microtubules help move cargo around inside cells. Chromosomes, which contain the cell's genetic blueprints, are the microtubule's most precious cargo. Before a cell divides, microtubules grow from the ends of the dividing cell towards the middle, where they attach to the chromosomes that are lined up along the centerline. Then the microtubules shrink and drag the chromosomes back to the opposite ends of the cell. This allows each of the new cells to get one copy of each chromosome.

When the microtubules are growing, a molecule called guanosine triphosphate (or GTP) is attached to the proteins at the end of the filament. This acts like a cap and protects the microtubule from shrinking. Later a chemical reaction converts GTP into GDP (short for guanosine diphosphate). Without the protective GTP cap, the microtubule quickly shrinks. At the same time, the proteins that make up the microtubule also change shape. In the microtubule, the proteins adopt a straight shape when GTP is attached. The proteins favor a different shape in the microtubule when GDP is attached. However, it is unclear if or how these shape changes contribute to how a microtubule grows or shrinks.

Geyer et al. now show how this shape shifting can influence microtubule shrinking, by first identifying a mutation in yeast microtubule proteins that cause the proteins to remain straight even when GDP is attached. Next, powerful microscopes were used to make time-lapse videos of the mutated microtubules. This allowed Geyer et al. to observe how the mutated microtubules behaved and compare this to the behavior of normal microtubules.

The experiments revealed that the mutated microtubules were less likely to begin shrinking than typical microtubules. The mutated microtubules also shrunk more slowly. These findings indicate that the shape changes control the speed of shrinking and frequency of entering the shrinking phase. These new details about the control of microtubule growth and shrinkage may help scientists studying how cell division happens in both healthy and cancerous cells.

example of selective interactions with specific conformations of αβ-tubulin comes from work in our laboratory showing that TOG domains from the microtubule polymerase Stu2p bind preferentially to the curved conformation of αβ-tubulin (*Ayaz et al., 2012*; *2014*), and that this preference of TOG domains for curved αβ-tubulin is probably what allows the polymerase to localize to the growing tip of the microtubule (*Ayaz et al., 2014*). These and numerous studies of other regulatory proteins (*Desai et al., 1999*; *Gigant et al., 2000*; *Peters et al., 2010*; *Alushin et al., 2010*; *Bechstedt et al., 2014*) contribute to the emerging view that selective binding to distinct conformations of αβ-tubulin represents a common and important strategy for recognizing and controlling microtubules.

How the αβ-tubulin conformational cycle contributes to microtubule dynamics is less understood. Indeed, even though structural rearrangements in αβ-tubulin have long been known to occur during and after microtubule incorporation (*Mandelkow et al., 1991b*; *Chrétien et al., 1995*; *Hyman et al., 1995*), actually showing that these rearrangements contribute to and are required for the dynamic properties of microtubules remains challenging (*Rice et al., 2008*; *Wang and Nogales, 2005*; *Buey et al., 2006*; *Jánosi et al., 2002*; *Molodtsov et al., 2005*). For example, GTP-bound αβ-tubulin 'straightens' during the formation of longitudinal and lateral lattice contacts (*Buey et al., 2006*; *Rice et al., 2008*; *Nawrotek et al., 2011*), but how (or even if) the energetic cost of these assembly-dependent conformational changes affects polymerization dynamics remains debated. A similar argument can be made about the αβ-tubulin conformational changes that occur as a consequence of GTP hydrolysis deeper in the microtubule lattice. One reason these questions are still debated is that to date in vitro microtubule dynamics have only been measured for a small number of αβ-tubulin mutants (*Gupta et al., 2002*; *Dougherty et al., 2001*; *Sage et al., 1995*), and it has not yet been possible to link altered dynamics to a change in the αβ-tubulin conformational cycle.

In the present study we sought to discover how multiple αβ-tubulin conformations affect microtubule dynamics by identifying mutations that selectively perturb some aspect of the conformational cycle. We reasoned that mutating buried residues with different packing environments in the curved and straight conformations might provide a way to perturb the conformational cycle. We focused on helix H7 because it is an element that is positioned differently in the 'curved' (unpolymerized) and

'straight' (polymerized) conformations of αβ-tubulin (*Nogales et al., 1998*; *Ravelli et al., 2004*) (*Figure 1A*), and it may act as a structural relay connecting the 'top' and 'bottom' of the tubulin sub-units (*Amos, 2004*). The equivalent helix in α-tubulin was also recently observed to move in response to GTP hydrolysis in the lattice (*Alushin et al., 2014*). We chose to study the T238A mutation in β-tubulin because this sidechain resides on helix H7 and is buried in the core of the protein (*Figure 1A*), and because prior studies (*Thomas et al., 1985*; *Machin et al., 1995*) showed that the

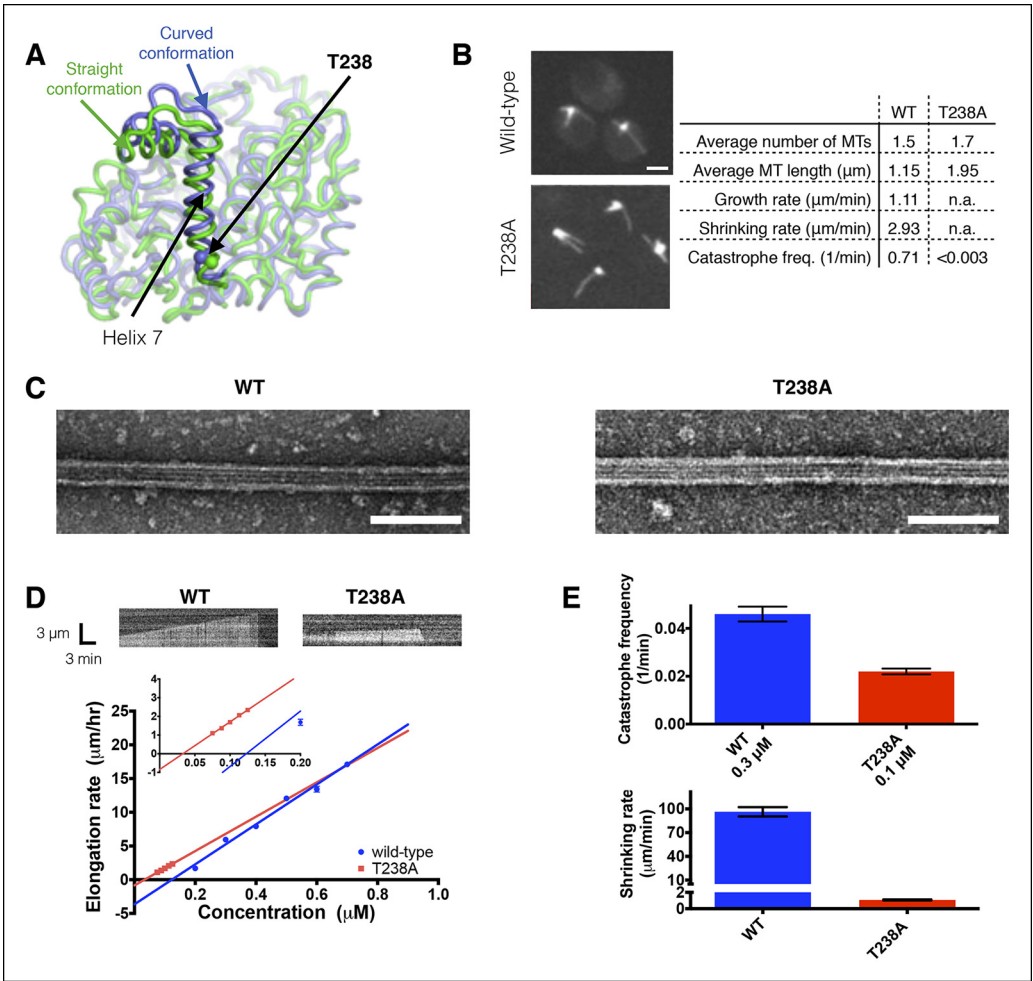

**Figure 1.** αβ-tubulin containing a buried mutation in β-tubulin gives hyperstable microtubules in vivo and in vitro. (A) Superposition of polymerized ('straight', green, PDB 1JFF) and unpolymerized ('curved', blue, PDB 4FFB) conformations of β-tubulin. T238 is solvent inaccessible and resides on a helix (H7) that undergoes a piston-like movement between the two conformations. The view of β-tubulin shown here is as if from the center of the microtubule looking out, with the plus end at the top. (B) Time-lapse imaging of live yeast shows that cells expressing T238A αβ-tubulin make longer, less dynamic microtubules than their wild-type (WT) counterparts. N = 13, 18 for WT and T238A microtubules respectively, with t = 35, 72 min total time observed. See also *Videos 1,2*. Bar = 2 µm (C) Negative stain electron micrographs (magnification: 23,000x) of wild-type (left) and β: T238A (right) microtubules. Mutant microtubules have normal structure. Bar = 100 nm. (D) In vitro, T238A and wild-type microtubules show similar concentration-dependent growth rates (slopes: 29.6 +/- 1.8 µm/hr/µM for wild-type, 25.5 +/- 0.8 µm/hr/µM for β:T238A, the differences between the slopes are not statistically significant; x-intercepts: 0.12 µM for wild-type, 0.033 µM for β:T238A, this difference is statistically significant; significance of differences in regression parameters was evaluated using GraphPad Prism). Representative kymographs are shown above the plots. (N = 12 for all points except 0.2 µM where N = 20; bars show s.e.m.) (E) T238A microtubules catastrophe less frequently than wild-type (top; N = 99, 115 for wild-type and T238A respectively, bars show s.d.) and show a ~hundredfold slower rate of post-catastrophe shrinking (bottom; N = 16 for wild-type and T238A, bars show s.e.m.).

β:T238A phenotype was consistent with hyperstable microtubules. The molecular mechanism by which the β:T238A mutation affects microtubule dynamics is unknown, in part because the polymerization dynamics of β:T238A αβ-tubulin have not been studied in vivo or in vitro. Our studies now reveal that the β:T238A mutation affects microtubule dynamics by dramatically reducing the frequency of catastrophe and the rate of shrinking, and that the mutation affects microtubule structure by attenuating post-GTPase conformational changes in the lattice. The effects of the mutation are reminiscent of the way the microtubule stabilizing drug taxol affects microtubule structure (*Alushin et al., 2014*). By showing that the β:T238A mutation weakens the coupling between the conformational and nucleotide cycles, our results demonstrate that catastrophe can occur despite diminished post-GTPase conformational changes in the microtubule lattice, and that the strength of the allosteric response to GDP dictates the rate of microtubule shrinking and the frequency of microtubule catastrophe.

## Results

### The buried T238A mutation in β-tubulin hyperstablizes microtubules in vivo and in vitro

The β:T238A mutation was previously identified by virtue of drug- and temperature-sensitive phenotypes that were consistent with hyperstable microtubules (*Thomas et al., 1985*). However, prior studies of this mutant have not directly examined its polymerization dynamics in vivo or in vitro (*Machin et al., 1995*; *Dorn et al., 2005*). To obtain insight into how the mutation affects polymerization dynamics in cells, we used time lapse imaging in GFP-Tub1p expressing strains to measure microtubule dynamics of wild-type and β:T238A-containing yeast. These measurements revealed that β:T238A-tubulin forms static microtubules (neither growing nor shrinking) that are on average over 50% longer than the dynamic microtubules that form in a strain with wild-type β-tubulin (*Figure 1B* and *Videos 1,2*). Because the mutated site is solvent inaccessible, this striking change in microtubule dynamics cannot be the result of a direct perturbation of a polymerization interface or of an interaction with one or more regulatory proteins.

To determine how the buried β:T238A mutation affected microtubule dynamics in vitro, we purified β:T238A αβ-tubulin from an overexpressing strain of yeast (*Johnson et al., 2011*) and used time-lapse differential interference contrast microscopy to measure its polymerization dynamics. We were unable to measure mutant and wild-type microtubule dynamics at equivalent concentrations, because β:T238A αβ-tubulin showed abundant spontaneous nucleation at the higher concentrations where we measured wild-type, and wild-type αβ-tubulin does not elongate measurably at the low concentrations where we were able to measure β:T238A dynamics without excessive nucleation. Mutant and wild-type microtubules nevertheless show similar concentration-dependent elongation rates: fitting lines to mutant and wild-type data reveals that the x-intercepts of the two datasets

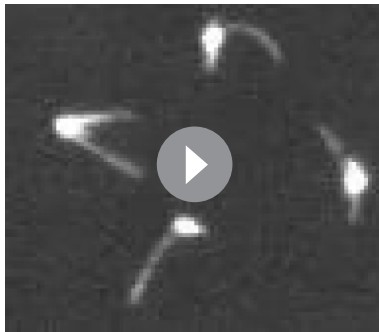

**Video 1.** Microtubule dynamics in wild type yeast. Time-lapse images of Tub1-GFP in *TUB2* (β-tubulin) cells were taken at 15-s intervals; video plays at 4 frames/s. A frame from this video is shown in *Figure 1B*.

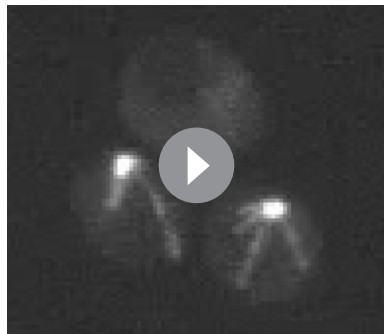

**Video 2.** Microtubule dynamics in *tub2-150* (T238A mutation in Tub2p) yeast. Time-lapse images of Tub1-GFP in *tub2-150* cells were taken at 15-s intervals; video plays at 4 frames/s. A frame from this video is shown in *Figure 1B*.

(0.12 and 0.033 µM for wild-type and β:T238A, respectively) differ by a factor of ~3.5 and that the difference in slope (29.6 and 25.5 µm/hr/µM for wild-type and β:T238A, respectively) is not statistically significant (*Figure 1D*). Because the x-intercept and slope respectively relate to the apparent affinity and association rate constant for elongation, our data indicate that the mutation has little effect on the apparent biochemistry of microtubule elongation. Consistent with this biochemical similarity, negative stain electron microscopy revealed that mutant and wild-type microtubules show similar structure (*Figure 1C*). In striking contrast to the shared elongation behavior, after catastrophe β:T238A microtubules shrink roughly hundredfold more slowly than wild-type (1.1 µm/min for β: T238A compared to 96 µm/min for wild-type, *Figure 1E*, bottom). Thus, the mutation significantly strengthens the lattice contacts that dictate the rate of microtubule shrinking. Finally, β:T238A microtubules also undergo catastrophe much less frequently than wild-type. The lower catastrophe frequency we observed is especially notable when considering that in these assays the T238A microtubules were growing much slower than wild-type because of the ~threefold lower concentration of αβ-tubulin used for the mutant (*Figure 1E*, top).

## Mutant-induced changes in polymerization dynamics do not result from defective GTPase activity

The β:T238A mutation stimulated spontaneous nucleation and reduced the frequency of catastrophe and the rate of shrinking, all without substantially affecting elongation. It seemed possible that a defective GTPase cycle might explain these observations. We reasoned that if the increased spontaneous nucleation of the β:T238A mutant resulted from slower/defective GTPase activity, then both mutant and wild-type should nucleate with similar efficiency when GTP hydrolysis cannot occur. We initially attempted to use GMPCPP, the hydrolysis-resistant nucleotide of choice for vertebrate microtubules (*Hyman et al., 1992*), but GMPCPP did not support elongation of yeast microtubules in our dynamics assays. Yeast microtubules polymerized readily in the presence of GTPγS, however, indicating that GTPγS better mimics GTP for yeast microtubules. We observed that even in the presence of GTPγS, wild-type microtubules show substantially less nucleation than T238A microtubules (*Figure 2A,B*). Thus, the abundant nucleation from the mutant cannot be ascribed to a defect in GTPase activity. Instead, the mutation must be affecting some other property that limits spontaneous nucleation in wild-type αβ-tubulin.

To support the idea that the mutation-induced changes in dynamics result from something other than a defect in GTPase activity, we assayed the nucleotide content of wild-type and β:T238A microtubules. We allowed wild-type and β:T238A αβ-tubulin to spontaneously polymerize at 1 or 2 µM concentration (within the range of concentrations tested in *Figure 2A*) in the presence of $^{32}$P-GTP. We harvested the microtubules by centrifugation, denatured them to release bound nucleotides, and analyzed the nucleotide content at the exchangeable site using thin layer chromatography (TLC) (*Figure 2C,D*). We probed for a GTPase defect using α-labeled nucleotide to measure the amounts of GTP and GDP in microtubules, and for a phosphate release defect using γ-labeled nucleotide to measure the amounts of GTP and Pi in microtubules. To avoid possible complications that might arise from the very different catastrophe frequencies and shrinking rates, we initially performed these assays under 'catastrophe free' and slow shrinking conditions by using (unlabeled) GTPγS to support assembly. These experiments revealed that wild-type and mutant microtubules contain similarly low amounts of GTP (2–4% of total exchangeable nucleotide) and Pi (fewer than 3% of exchangeable sites) (*Figure 2C,D*). When we performed these assays using (unlabeled) GTP to support microtubule assembly, β:T238A microtubules contained similarly low amounts of GTP and Pi as we observed with GTPγS but wild-type microtubules appeared to contain a substantially greater amount (~50%) of GTP (*Figure 2C,D*). Yeast microtubules have previously been reported to contain more GTP than vertebrate microtubules (*Dougherty et al., 1998*). In control experiments (not shown) we confirmed that vertebrate microtubules contain very little GTP and that individual heterodimers were not pelleting. Because we observed very fast shrinking (*Figure 1*), little $^{32}$P-GTP in microtubules grown with GTPγS (*Figure 2C*), and normal plus-end recognition by an EB1 family protein (see below), it seems unlikely that wild-type yeast microtubules contain substantial amounts of GTP. The GTP containing material in our assay might instead reflect oligomers that form more readily for yeast αβ-tubulin. Whatever the mechanism, these experiments demonstrate that β:T238A αβ-tubulin remains competent to hydrolyze GTP, and that the altered nucleation and shrinking behavior cannot be ascribed to an accumulation of GTP or GDP.Pi that might result from a defect in

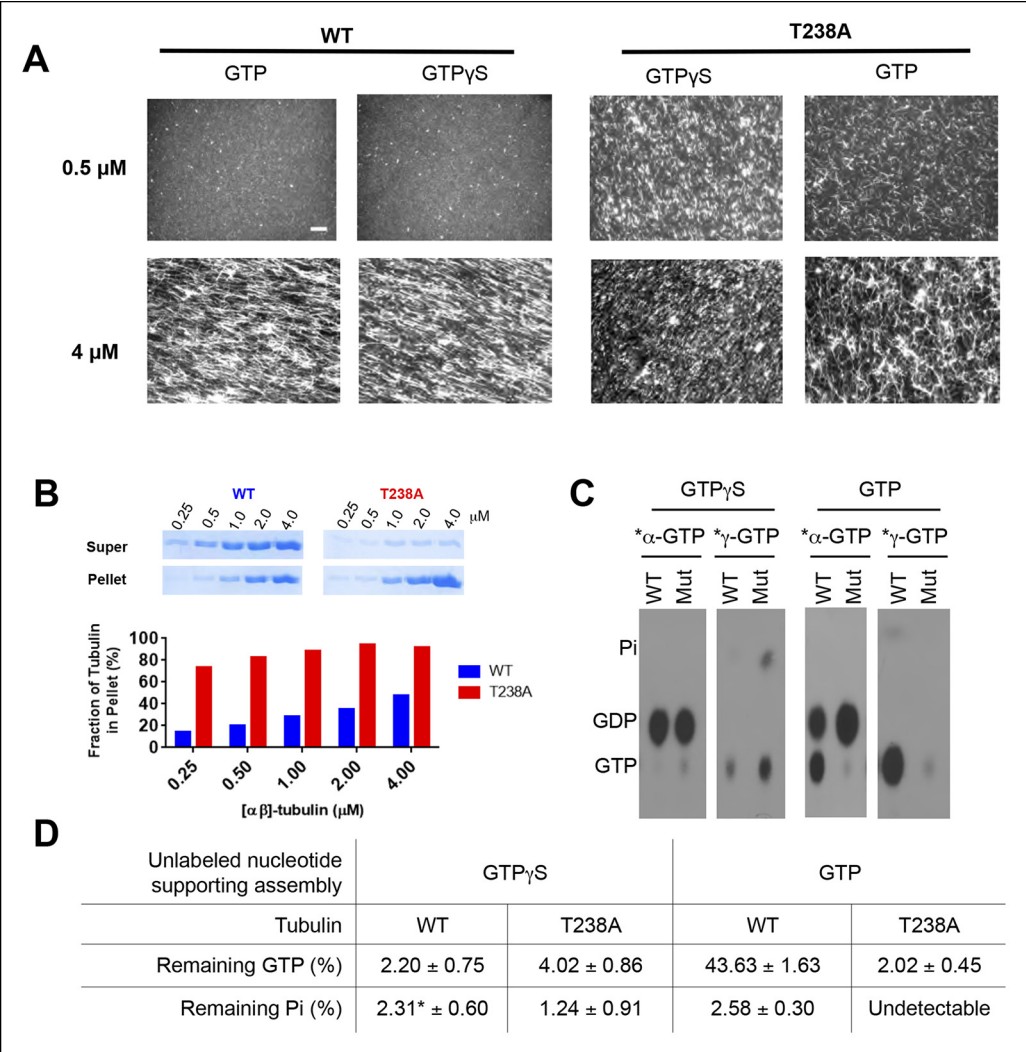

**Figure 2.** T238A $\alpha\beta$-tubulin undergoes spontaneous nucleation more readily than WT, even in the presence of a nonhydrolyzable GTP analog, GTPγS. (**A**) Fluorescent images of crosslinked microtubules from spontaneous nucleation reactions. Even at low concentrations and in the presence of GTPγS, T238A tubulin shows increased spontaneous nucleation compared to WT. GTPγS reactions are presented next to each other to facilitate a side-by-side comparison. Scale bar in top left is 5 µm. (**B**) Microtubule spindown reactions show that under the same concentration range, in the presence of GTPγS, T238A tubulin produces a greater proportion of microtubules which sediment into the pellet. Gel images of supernatant and pellet fractions (top). (**C**) T238A microtubules do not accumulate GTP or GDP.Pi compared to wild-type. Images show TLC analysis of exchangeable nucleotide content of microtubules grown with GTP or GTPγS. Microtubules were spontaneously assembled using higher concentrations than for the dynamics assays: wild-type microtubules were prepared at 2 µM with GTP and 1 µM with GTPγS. β:T238A microtubules were prepared at 1 µM with either nucleotide. (**D**) Quantification of TLC data, n = 3 and errors shown represent s.d. *indicates one condition where we could not detect any Pi in one of the replicates; instead of using 0 we used the lower of the two other trials.

assembly-dependent GTPase activity or phosphate release. Some other, nonenzymatic mechanism must therefore account for the observed changes in polymerization dynamics.

## The buried T238A mutation in β-tubulin does not detectably 'straighten' unoligomerized αβ-tubulin

Unoligomerized GTP-bound αβ-tubulin adopts a curved conformation that is not compatible with the straight microtubule lattice (*Nawrotek et al., 2011*; *Pecqueur et al., 2012*; *Ayaz et al., 2012*;

*2014*). It seemed possible that the β:T238A mutation might stabilize microtubules by shifting the conformational preference of unpolymerized αβ-tubulin to favor a straight(er) conformation that is more compatible with the microtubule lattice. We used quantitative TOG binding assays (*Ayaz et al., 2014*) to determine if the β:T238A mutation changed the 'curvature' of unpolymerized αβ-tubulin. The rationale for this approach is based on the fact that TOG domains bind tightly to curved αβ-tubulin but very weakly to straight (*Figure 3A*) (*Ayaz et al., 2012*; *2014* ). Accordingly, β:T238A αβ-tubulin should bind less tightly to a TOG domain if the β:T238A mutation appreciably changes the conformation of αβ-tubulin, for example by straightening it. TOG1 and TOG2, two different TOG domains from the yeast microtubule regulatory factor Stu2p (*Wang and Huffaker, 1997*), show very similar affinity (within a factor of two) for unpolymerized T238A αβ-tubulin as they do for wild-type (*Figure 3B*). In energetic terms, this modest difference in affinity is roughly equivalent to one hydrogen bond. While we cannot state that the mutant and wild-type αβ-tubulin adopt identical conformations, their very similar TOG binding properties indicate that any differences in conformation must be small. Thus, we conclude that the β:T238A mutation does not significantly change the conformation of unpolymerized αβ-tubulin.

## The β:T238A mutation suppresses GTPase-dependent conformational changes in the microtubule lattice

Recent cryo-EM studies of microtubule structure provided atomic models for distinct microtubule lattices containing GTP, GTPγS, or GDP, with GTP favoring an 'expanded' form and GTPγS and GDP favoring a 'compacted' form (*Alushin et al., 2014*; *Zhang et al., 2015*). Plus-end tracking proteins in the EB1 family have been shown to discriminate between these GTP-like and GDP-like lattices (*Maurer et al., 2011*; *2012*; *2014*; *Zanic et al., 2009*). We used Bim1p (*Schwartz et al., 1997*), the yeast EB1 protein, to investigate if the β:T238A mutation affected αβ-tubulin conformation in the lattice.

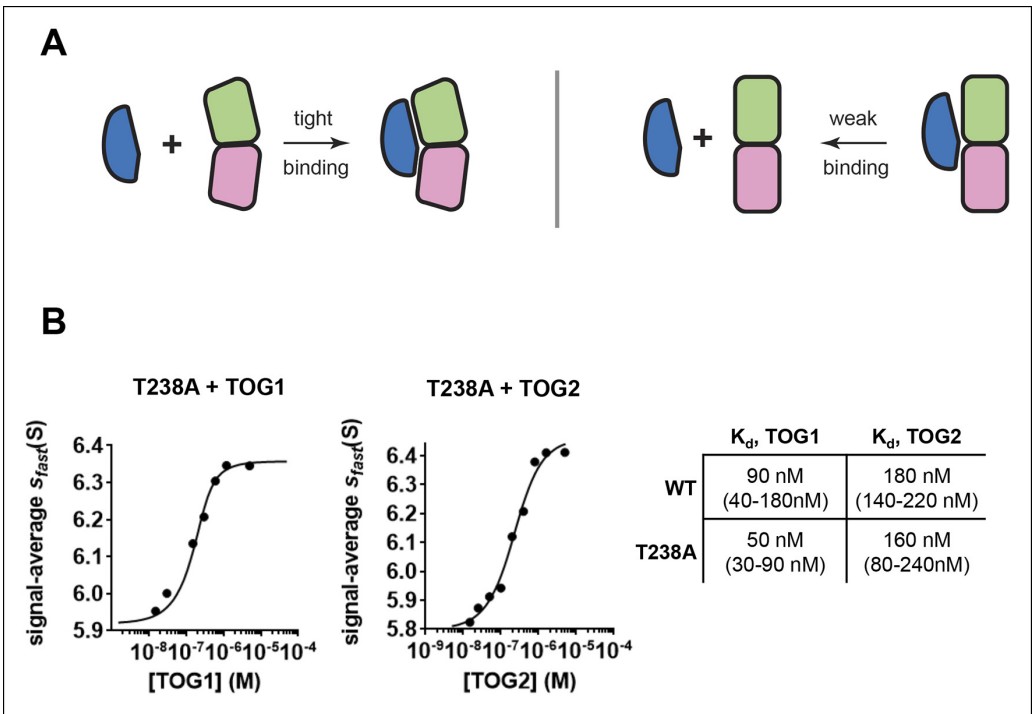

**Figure 3.** The buried *β*:T238A does not appreciably straighten unpolymerized *αβ*-tubulin. (**A**) Cartoon schematics illustrating that TOG domains (blue) bind tightly to curved αβ-tubulin (pink and green; left) but weakly to straight αβ-tubulin (right). (**B**) Isotherms for individual Stu2 TOG domains binding to T238A αβ-tubulin. Both TOG domains bind with comparable affinity to T238A and WT tubulin. 1 Σσ confidence intervals for fitted affinities are provided in parentheses.

Control experiments revealed that a Bim1-GFP fusion protein tracked the growing end of yeast microtubules, and that Bim1-GFP coated the entire length of GTPγS-containing yeast microtubules with cap-like intensity (*Figure 4*). We therefore infer that like other EB1 proteins on vertebrate microtubules (*Zanic et al., 2009*; *Maurer et al., 2011*), Bim1 discriminates between GTP and GDP forms of the yeast microtubule lattice (presumably, expanded and compacted). In marked contrast to its behavior on wild-type microtubules, at 50 nM concentration Bim1-GFP coated the entire length of 'dynamic' β:T238A microtubules with cap-like intensity. At lower concentrations (5 nM), the Bim1-GFP coat on the body of β:T238A microtubules is slightly weaker than at the cap but still substantially more intense than on wild-type (*Figure 4B*). This stronger coating reflects tighter Bim1 binding to mutant microtubules, and it occurs in spite of the fact that the mutant microtubules contain very little GTP or GDP.Pi in the exchangeable site (*Figure 2*). Thus, the β:T238A mutation attenuates the conformational response to GTP hydrolysis in the microtubule lattice. The structural consequence of the mutation resembles the effect of taxol binding, which also promotes an expanded conformation of αβ-tubulin in a GDP lattice (*Alushin et al., 2014*). Simply put, the mutation appears to have substantially uncoupled the conformational cycle from the nucleotide cycle in the lattice, allowing αβ-tubulin to retain GTP-lattice-like character even in a GDP lattice.

## The β:T238A mutation affects αβ-tubulin curvature on the microtubule end

The Bim1 experiments described above do not report on the conformation of αβ-tubulin at the very microtubule end because EB1 proteins do not bind there (*Maurer et al., 2012*; *2014*). After fortuitously discovering that isolated TOG domains from the microtubule polymerase Stu2p stimulate the depolymerization of stabilized microtubules in a dose-dependent way, it seemed that TOG domains might provide an alternative way to probe the conformation of αβ-tubulin on the microtubule end. In light of the fact these TOG domains bind preferentially to the curved conformation of αβ-tubulin, we reasoned that the underlying cause of this induced depolymerization was TOG-mediated stabilization of the curved and faster dissociating conformation of αβ-tubulin on the microtubule end (*Figure 5*). According to this view, TOG-induced depolymerization thus provides an assay that probes the linkage (or lack thereof) between changes in αβ-tubulin conformation on the polymer end and microtubule shrinking rate.

Using drug-stabilized microtubules as a substrate, the rate of TOG-induced depolymerization increases linearly over at least a 20-fold range of TOG concentration. This linear concentration-dependence is consistent with a collisional mechanism in which the rate-limiting step is a TOG domain arriving at the microtubule end. It is not consistent with a mechanism in which the rate-limiting step is some kind of slow αβ-tubulin conformational change. Drug-stabilized β:T238A microtubules depolymerize substantially slower than wild-type at the same concentration of TOG domain (*Figure 5*). Only at higher TOG concentrations do β:T238A microtubules show appreciable dose-dependent TOG-induced depolymerization (*Figure 5*). Similar results were obtained using MTs stabilized with GTPγS (*Figure 5*).

TOG domains bind with comparable affinity to curved wild-type and mutant αβ-tubulin, so differences in TOG affinity cannot explain the markedly slower TOG-induced shrinking of β:T238A microtubules. Differences in the amount of curved αβ-tubulin at wild-type and mutant microtubule ends could explain the observed differences in TOG-induced shrinking. In principle there might be other mechanisms that strengthen association without affecting the propensity to be curved on the microtubule end. However, while such 'curvature invariant' mechanisms in αβ-tubulin might explain the slow post-catastrophe shrinking of the mutant, the conformation-selective nature of TOG:tubulin interactions means that this alternative view is not easily reconciled with the observed differences in TOG-induced depolymerization. Thus, a mutant-induced change in the propensity to become/remain straight on the microtubule end is the simplest way to explain both the slower post-catastrophe shrinking and weaker TOG-induced depolymerization for the mutant. That the conformation of end-bound αβ-tubulins is affected by the β:T238A mutation is also consistent with the Bim1 coating, which demonstrated that the mutation affected αβ-tubulin conformation elsewhere in the lattice.

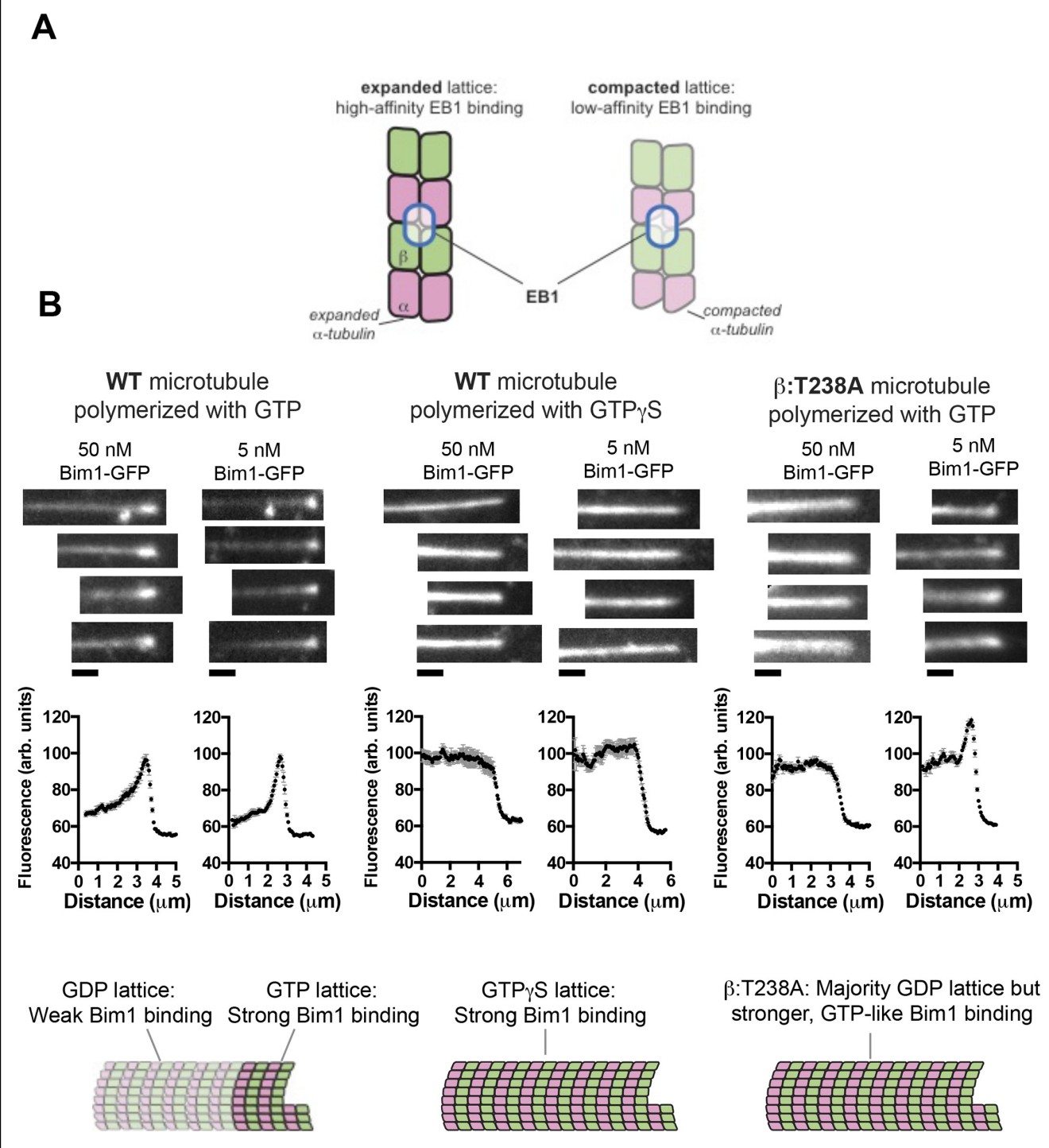

**Figure 4.** The buried β:T238A attenuates the conformational response to GDP in the lattice. (A) Cartoon schematics illustrating the basis of the assay: EB1 proteins (blue outlined ovals) bind tightly to the GTP-lattice conformation of αβ-tubulin (left) but weakly to the GDP-like conformation (right). (B) Images of the distribution of Bim1-GFP (an EB1 family protein) on wild-type (left and center) or mutant microtubules (right). 4 microtubules are shown for each condition, and Bim1-GFP was present at 50 nM and 5 nM concentration for all microtubules. Plots represent average Bim1-GFP intensity on n = 9 microtubules for each condition. Error bars represent s.e.m. Cartoons illustrate the likely lattice conformation inferred from the Bim1-GFP binding. β:T238A microtubules appear GTP-like even though they contain GDP. Scale bars: 1 μm.

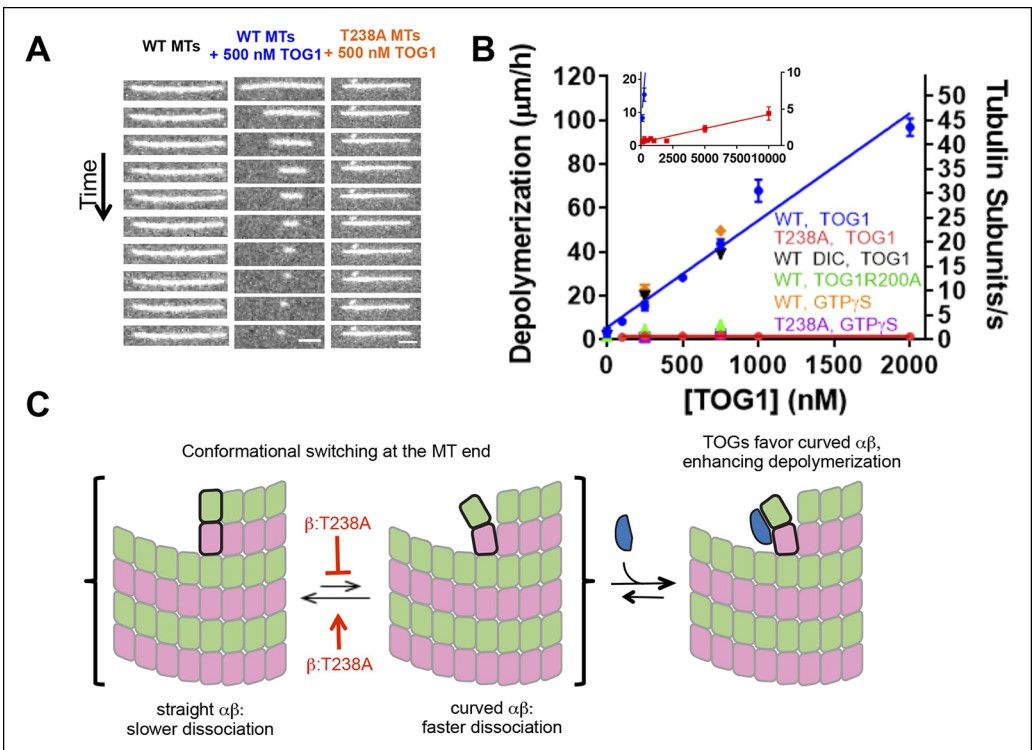

**Figure 5.** β:T238A microtubules are resistant to TOG-induced depolymerization. (**A**) A time series of images of stabilized fluorescent microtubules show that under the same concentration of TOG1, T238A microtubules depolymerize substantially slower than WT microtubules. Scale bars: 1 μm. (**B**) Quantification of the dose-dependence of the rate of TOG-induced depolymerization. Inset: a plot showing that under higher TOG1 concentrations, T238A microtubules also undergo induced depolymerization. We observed similar rates of TOG-induced depolymerization using DIC instead of fluorescence to monitor microtubule length (black triangles), as well as using GTPγS instead of epothilone as a stabilizing reagent (orange diamonds, purple squares). Additional control experiments demonstrate that the TOG-induced depolymerization is greatly reduced when a weakly-binding TOG1 mutant (R200A) is used (green triangles). N = 30 for fluorescence measurements of wild-type TOG1-induced depolymerization of wild-type or β:T238A microtubules, N = 20 for the rest. Error bars represent s. e.m. (**C**) Cartoon model illustrating the mechanism of TOG-induced depolymerization, resulting from TOG-stabilization of tubulin subunits sampling curved conformation at the end of the microtubule. T238A microtubules depolymerize slower due to a decrease in tubulin subunits sampling the curved conformation once incorporated into the polymer.

## What is the structural origin of the β:T238A effects?

To gain insight into the local interactions responsible for the β:T238A effects on microtubule dynamics, we measured benomyl sensitivity/resistance and in vivo microtubule dynamics for serine and valine substitutions at the same position (*Figure 6*). Yeast expressing β:T238S αβ-tubulin are comparably benomyl resistant to β:T238A, but significantly less benomyl dependent (*Figure 6A*). By contrast, yeast expressing β:T238V αβ-tubulin are substantially less resistant to benomyl, and show no benomyl dependence (*Figure 6A*). Measurements of microtubule dynamics in vivo are consistent with the benomyl phenotypes: β:T238S microtubules are β:T238A-like (static) whereas β:T238V microtubules are more wild-type-like (dynamic but with somewhat slower shrinking) (*Figure 6B*). The trend with sidechain size and the strong effect obtained from the Thr to Ser mutation at position 238 suggest that some form of 'steric overpacking', not hydrogen bonding, contributes to destabilize the straight, expanded conformation of αβ-tubulin in microtubules containing GDP.

Reducing the size of β:C354, a buried sidechain that packs against β:T238 (*Figure 6C*), has also been shown to stabilize microtubules (*Gupta et al., 2001*; *2002*). Indeed, β:C354A or β:C354S substitutions dramatically reduced the rate of microtubule shrinking and the frequency of catastrophe (*Gupta et al., 2002*). If these volume-reducing mutations also stabilize the expanded conformation

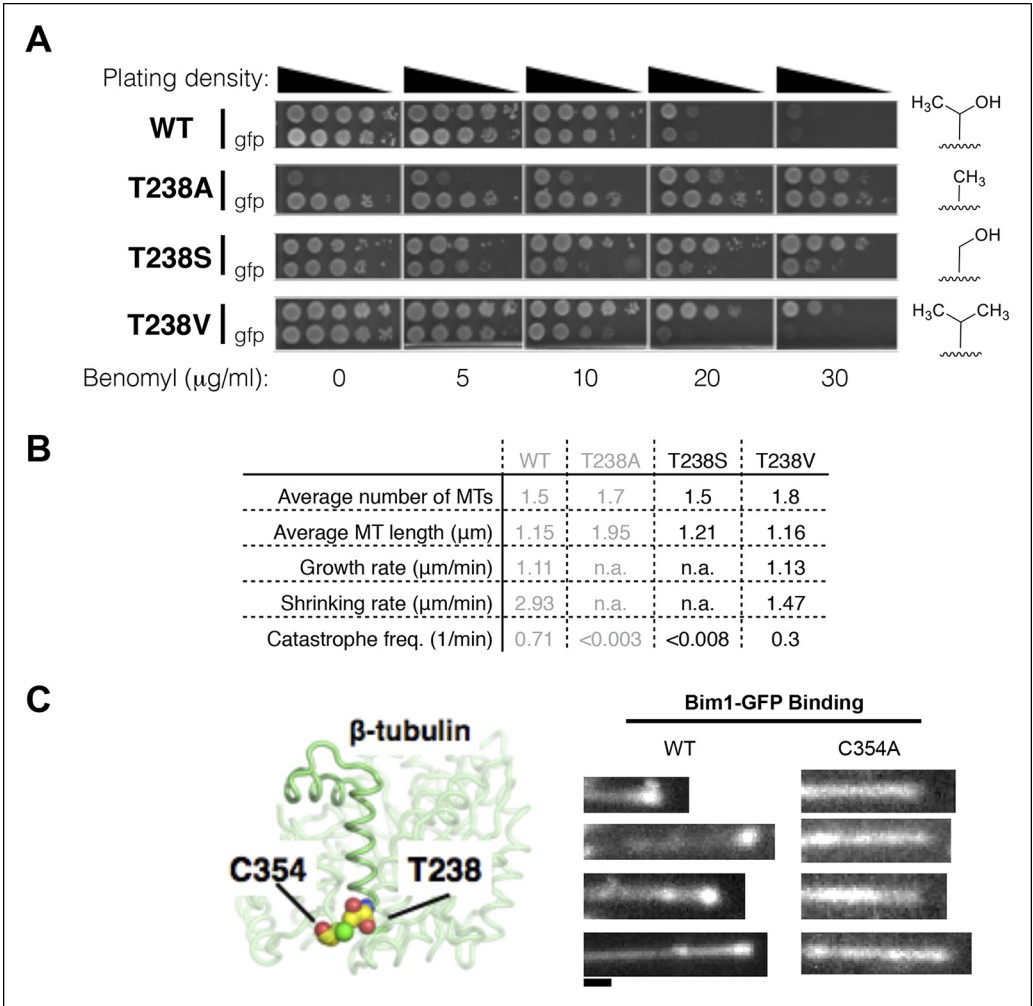

**Figure 6.** Insights into the mechanism underlying the β:T238A effects on the conformational cycle. (**A**) Yeast with different substitutions for T238 (for each mutant, strains with and without GFP-Tub1p are shown) show intermediate degrees of benomyl resistance/dependence. The volume of the packing defect may be related to the magnitude of the resulting phenotype: T238S has a phenotype closer to T238A whereas T238V has a phenotype closer to wild type. (**B**) In vivo microtubule dynamics of yeast containing T238S and T238V αβ-tubulin. N = 24, 14 for T238S and T238V microtubules respectively, with t = 120, 43 min total time observed. (**C**) The buried β: C354A mutation immediately proximal to T238 (left) has also been shown to give slowly shrinking microtubules, and also shows an expanded lattice (right). The view of the structure is as if from the center of the microtubule looking out, with the plus end at the top. Scale bar: 1 μm.

of αβ-tubulin in a GDP lattice, it would provide complementary support for our overpacking-based rationale for the β:T238(A, S, V) effects. We purified β:C354A αβ-tubulin and used Bim1-GFP to probe its conformation in microtubules. β:C354A microtubules showed Bim1-GFP coating similar to what we observed for β:T238A, indicating the two mutations stabilize microtubules through a common conformational mechanism (*Figure 6C*). Positions 238 and 354 in β-tubulin show strong evolutionarily conservation (yeast numbering; β:238 is 65% T and 34% C whereas β:354 is 99% C), consistent with an important functional role for these residues. Together, the β:T238A and β:C354A results suggest that changes in packing volume near helix H7 can dictate the dynamic properties of microtubules by tuning an allosteric response to GDP.

## Discussion

We demonstrate that relatively conservative buried mutations in β-tubulin can diminish the conformational response to GDP in the lattice, and that this altered structural response yields dramatic effects on microtubule dynamics. By discovering that mutations can uncouple the conformational and nucleotide cycles of αβ-tubulin, our findings provide new insights into ways that the αβ-tubulin conformational cycle dictates microtubule polymerization dynamics (*Figure 7*).

Our experiments using the EB1 family protein Bim1 showed that in the lattice a major effect of the β:T238A mutation is to substantially suppress conformational changes that normally occur as a consequence of GTPase activity (*Hyman et al., 1995*; *Alushin et al., 2014*; *Zhang et al., 2015*). The different responses of yeast and vertebrate microtubules to GMPCPP, and the lack of high-resolution structures for wild-type or mutant yeast microtubules, make it difficult to know the specific conformations involved and how they relate to the nucleotide cycle. However, the stronger binding of Bim1 to GDP-containing β:T238A microtubules indicates that the mutant microtubule lattice retains more GTP-like character than does wild-type. This observation about αβ-tubulin in the body of the microtubule cannot by itself explain the slow shrinking, however, because the rate of shrinking is determined by the properties of incompletely surrounded terminal subunits, on which EB1 does not report. Indeed, understanding how different lattice structures relate to dynamic properties of microtubules remains a significant challenge. Our experiments with TOG domains indirectly probed the conformation of terminal subunits and showed that the β:T238A mutation reduces the propensity of terminal αβ-tubulins to be curved. Thus, the slow shrinking and lower frequency of catastrophe in the mutant most likely result from the attenuated allosteric response to GDP; this in turn results in a straighter and more strongly associated GTP-like conformation on the microtubule end, despite having GDP at the longitudinal interface (*Figure 7B*).

It is remarkable that the majority of the mutation-induced effects are confined to post-GTPase conformational changes in the lattice. Not observing significant effects on elongation is somewhat surprising, because it had been anticipated that the conformational changes required to enter the lattice would contribute to polymerization dynamics by opposing incorporation into the lattice. The lack of substantial change in the observed concentration-dependence of elongation rates could indicate that the β:T238A mutations do not affect the propensity for αβ-tubulin to 'straighten' on a GTP lattice. However, this is not consistent with increased spontaneous nucleation of β:T238A αβ-tubulin in the presence of GTPγS, which indicated that the mutation did affect conformational transitions that occur in/on a GTP lattice. It could be that we did not observe substantial effects on elongation because the energetics of curved to straight transitions on the microtubule end are not rate-contributing for elongation, perhaps because these transitions occur subsequent to end binding.

The significant mutant-induced structural changes in the lattice were not accompanied by equivalent changes in the conformation of unoligomerized αβ-tubulin. The differential response to the mutation inside and outside of microtubules (*Figure 7A,B*) is consistent with the view that the curved conformation represents the 'ground state' of αβ-tubulin independent of nucleotide state (*Rice et al., 2008*; *Buey et al., 2006*; *Nawrotek et al., 2011*; *Pecqueur et al., 2012*; *Ayaz et al., 2012*), that the nucleotide acts across the longitudinal interface (*Rice et al., 2008*; *Nawrotek et al., 2011*), and that nucleotide-dependent interactions with the microtubule lattice are what drive αβ-tubulin conformational transitions (*Rice et al., 2008*; *Buey et al., 2006*; *Nawrotek et al., 2011*). Indeed, the changes we observed can largely be explained by an impaired allosteric response to GDP in the lattice. It will be interesting to discover in future work if other mutations in β-tubulin can straighten unpolymerized αβ-tubulin or modulate the allosteric response(s) to nucleotide in the lattice, and to determine if mutations in α-tubulin can yield similar effects.

In summary, we showed that buried mutations of or near β:T238 alter the allosteric response to GDP in the microtubule lattice, with dramatic consequences for catastrophe frequency and shrinking rate. By describing microtubules with identical interface composition that nevertheless undergo strikingly different polymerization dynamics, our data demonstrate that allostery in the lattice dictates functionally important aspects of microtubule polymerization dynamics (*Figure 7C*). Based on the central role of allostery in controlling the frequency of catastrophe and the rate of shrinking, we speculate that cooperative conformational linkage in the lattice amplifies the individual response to nucleotide. Together, the allosteric response to GDP and the intrinsic bias of αβ-tubulin toward the curved conformation elevate and separate the threshold concentrations for persistent elongation

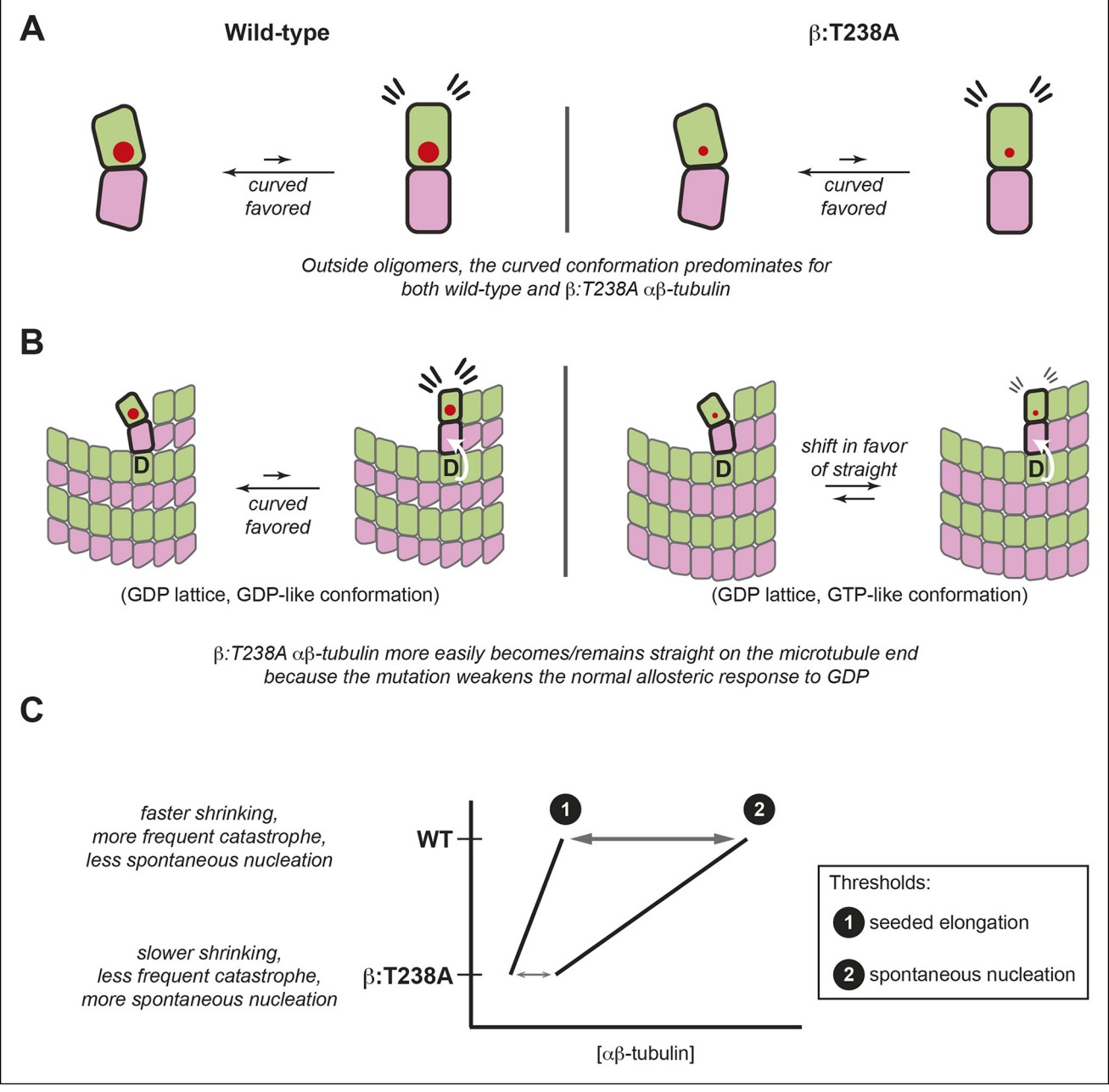

**Figure 7.** The αβ-tubulin conformational cycle and its impact on microtubule dynamics. (**A**) Unoligomerized wild-type and β:T238A αβ-tubulin both adopt the curved conformation. That the straight conformation is strained (flare marks) even with the mutation-induced reduction in packing volume (cartooned by larger and smaller red circles for wild-type and β:T238A, respectively) suggests that extrinsic factors like interactions with the lattice control straightening. (**B**) Compared to wild-type, β:T238A αβ-tubulin is better able to populate a straight conformation on the end of GDP containing microtubules. 'D' indicates GDP underneath the bolded terminal subunit, compaction is represented as in *Figure 4*, and the white arrow cartoons trans-acting nucleotide (see text). The increased ability of the mutant to be straight could result from an altered response to GDP on the longitudinal interface, from more favorable interactions with the expanded lattice, or from a combination of both. (**C**) The schematic phase diagram illustrates that the mutation-induced changes to the αβ-tubulin conformational cycle decrease the threshold concentrations for appreciable elongation against catastrophe (line 1) and for spontaneous nucleation (line 2), and also narrows the gap between them. The normal αβ-tubulin conformational cycle contributes to microtubule dynamics and makes them more amenable to regulation.

and for spontaneous nucleation (*Figure 7C*), and underlie the fast shrinking rate that makes catastrophe more decisive. These allosteric contributions to microtubule dynamics make microtubules more

amenable to regulation by cellular factors that enhance elongation, trigger catastrophe, and promote nucleation.

## Materials and methods

### Protein expression and purification

Plasmids to express TOG1, TOG2 and wild-type yeast αβ-tubulin were previously described (*Johnson et al., 2011*; *Ayaz et al., 2012*; *2014*). A Bim1-GFP construct, in pHAT vector containing C-terminal EGFP-tag followed by a *Strep*-tag II, was a gift from Dr. Gary Brouhard. A plasmid to express the T238A mutation of Tub2p (yeast β-tubulin) was made by QuikChange (Stratagene) mutagenesis, using an expression plasmid for wild-type Tub2 as template and with primers designed according to the manufacturer's instructions. A C-terminal FlAsH (*Griffin et al., 2000*) sequence was added to a Tub1 construct using polymerase chain reaction with primers designed to add the sequence WDCCPGCCK (*Griffin et al., 2000*). The integrity of all expression constructs was confirmed by DNA sequencing.

Wild-type or mutant yeast αβ-tubulin was purified from inducibly overexpressing strains of *S. cerevisiae* using Ni-affinity and ion exchange chromatography (*Johnson et al., 2011*; *Ayaz et al., 2012*; *2014*) with the exception that T238A and wild-type FlAsH mutants were eluted from the Ni-affinity column with 200 mM NaCl. Prior to ion exchange chromatography, T238A Ni elution fractions were treated with Universal Nuclease (Pierce) at RT for 1 hr. Tubulin samples were stored in storage buffer (10 mM PIPES pH 6.9, 1 mM $MgCl_2$, 1 mM EGTA) containing 20 or 50 µM GTP depending on the application. The TOG1(1-317) and TOG2(318-560) were expressed in E. coli with C-terminal $His_6$ tags and purified using Ni-affinity, gel filtration (TOG1) and ion exchange chromatography (TOG2) (*Ayaz et al., 2012*; *2014*). TOG domains were stored in RB100 (25 mM Tris pH 7.5, 100 mM NaCl, 1 mM $MgCl_2$, 1 mM EGTA). Expression of Bim1-GFP was induced in the BL21(DE3) strain of *E. coli* with 0.5 mM IPTG for 15 hrs at 16C. Single pellets were resuspended in 50 mM $Na_2HPO_4$, 300 mM NaCl, 40 mM imidazole and sonicated for 30 min in the presence of PMSF. Lysates were clarified by centrifugation. Cleared lysate was loaded onto a His60 Superflow Column (Clontech) and the final sample was eluted in 200 mM imidazole. Elution fractions were loaded onto a 3 mL Strep-Tactin Superflow column (IBA, Germany) and eluted in RB100 with 5 mM desthiobiotin.

### Yeast strains

For expression:
JEL1 MATα leu2 trp1 ura3−52 prb1−1122 pep4−3 Δhis3::PGAL10-GAL4
For analysis of growth phenotypes:
CUY 2172 MATa TUB2::URA3 ade2-101 his3-Δ200 leu2-1 lys2-801 ura3-52
CUY 2179 MATa tub2-T238A::URA3 ade2-101 his3-Δ200 leu2-1 lys2-801 ura3-52
CUY 2198 MATa tub2-T238S::URA3 ade2-101 his3-Δ200 leu2-1 lys2-801 ura3-52
CUY 2200 MATa tub2-T238V::URA3 ade2-101 his3-Δ200 leu2-1 lys2-801 ura3-52
For microtubule imaging:
CUY 2265 MATa TUB2::URA3 GFP-TUB1::LEU2::TUB1 ade2-101 his3-Δ200 leu2-1 lys2-801 ura3-52
CUY 2238 MATa tub2-T238A::URA3 GFP-TUB1::LEU2::TUB1 ade2-101 his3-Δ200 leu2-1 lys2-801 ura3-52
CUY 2208 MATa tub2-T238S::URA3 GFP-TUB1::LEU2::TUB1 ade2-101 his3-Δ200 leu2-1 lys2-801 ura3-52
CUY 2209 MATa tub2-T238V::URA3 GFP-TUB1::LEU2::TUB1 ade2-101 his3-Δ200 leu2-1 lys2-801 ura3-52

### In vivo experiments

*tub2* alleles were integrated into yeast as described previously (*Reijo et al., 1994*) in vivo imaging of microtubule dynamics was performed as described previously (*Huang and Huffaker, 2006*). Note that introduction of GFP-TUB1 into strains containing the *tub2-T238A* mutation lessens their dependence on benomyl for growth (*Figure 6*) and allowed us to image these cells in the absence of benomyl.

## Time-lapse measurements of microtubule dynamics

Flow chambers were prepared as described previously (*Gell et al., 2010*), with the exception that sea urchin axonemes (*Waterman-Storer, 2001*) were used to template yeast MT growth. Chambers were rinsed with BRB80 (80 mM PIPES pH 6.9, 1 mM MgCl$_2$, 1 mM EGTA), followed by 10 min incubation with sea urchin axonemes. Chambers were then blocked with 1% F-127 Pluronic in BRB80 for 5 min, and washed with 1X PEM (100 mM PIPES pH 6.9, 1 mM EGTA, 1 mM MgSO$_4$) containing 1 mM GTP. Reaction chambers were sealed with VALAP after addition of αβ-tubulin. Wild type or mutant yeast αβ-tubulin (in storage buffer containing 50 μM GTP) was taken from -80°C, rapidly thawed, and passed through a 0.1 μm centrifugal filter at 4°C to remove aggregates. The concentration of αβ-tubulin was measured by UV absorbance using an extinction coefficient of 115000 M$^{-1}$cm$^{-1}$. Protein was kept on wet ice for no more than 30 min before measuring MT dynamics. MT dynamics were imaged by differential interference contrast microscopy (DIC) using an Olympus IX81 microscope with a Plan Apo N 60x/1.42 NA objective lens and DIC prisms. Illumination at 550 nm was obtained by inserting a bandpass filter of 550/100 nm (Olympus) in the light path. Temperature was maintained at 30°C using a WeatherStation temperature controller with enclosure fit to the microscope's body. Micro-Manager 1.4.16 (*Edelstein et al., 2010*) was used to control the microscope and a Hamamatsu ORCA-Flash2.8 CMOS camera used to record the reactions. MT dynamics were recorded by taking an image every 500 ms for 1 to 2 hrs. At the end of each movie, a set of 100 out-of-focus background images was taken for background subtraction (see below). To improve signal to noise, batches of 10 raw images were averaged using ImageJ (*Schneider et al., 2012*) and intensity normalized before background subtraction. MT length was measured manually using a PointPicker plugin for ImageJ. Rates of MT elongation and catastrophe frequencies were determined as described previously (*Walker et al., 1988*).

## Electron microscopy

To prepare microtubules for electron microscopy, samples of wild-type (3-–4 μM) or β:T238A αβ-tubulin (2 μM) were prepared in 100 mM PIPES pH 6.9, 10% glycerol, 2 mM MgSO$_4$, 0.5 mM EGTA, 1 mM GTP and incubated at 30 °C for 1 hr. 5 μl of the assembly reactions were spotted onto freshly glow-discharged 400 mesh grids with a carbon coated formvar support (Ted Pella), incubated for 30 s, rinsed with water, and stained with 2% aqueous uranyl acetate. Negatively stained grids were imaged at 23,000 x magnification using a Tecnai G2 Spirit electron microscope equipped with a 2Kx2K CCD camera (Gatan).

## Assays for microtubule nucleation

Microtubule sedimentation assays were performed using a range of wild-type or mutant yeast αβ-tubulin concentrations (*Figure 2B*, 0.25 – 4 μM) in assembly buffer (100 mM PIPES pH 6.9, 10% glycerol, 2 mM MgSO$_4$, 0.5 mM EGTA) containing the indicated nucleotide. Samples were polymerized for 90 min at 30°C. For SDS-PAGE analysis, microtubules were pelleted by centrifugation at 60,000 rpm (~150000 x g) at 30°°C for 30 min in a pre-warmed TLA-100 rotor (Beckman-Coulter), supernatant was carefully removed, and the pellet was re-suspended in an equal volume of assembly buffer such that the pellet and supernatant fractions were of equal volume. To image the products of the nucleation reactions, the reactions were cross-linked by diluting 10-fold into assembly buffer containing 1% glutaraldehyde. Cross-linking was quenched after 3 min by 5-fold dilution into assembly buffer containing 20 mM Tris pH 6.8. 150 μL of the quenched, cross-linked reactions were applied to the top of a glycerol cushion (20% glycerol in BRB80) and spun through the cushion onto poly-L-lysine coated coverslips. Coverslips were washed with BRB80, fixed with methanol and stained using FITC-DM1α (Sigma-Aldrich) for imaging by epifluorescence, as described previously (*Ayaz et al., 2012*).

## Determining nucleotide content

Wild type (2 or 1 μM respectively for GTP or GTPγS) or β:T238A tubulin (1 μM for either nucleotide) was mixed with 100 μM GTP or GTPγS (containing 33 nM [α-$^{32}$P or γ-$^{32}$P]-GTP) in assembly buffer (see above) on ice. Microtubules were assembled and harvested by centrifugation at 80,000 rpm (~268000 x g) at 30°C for 10 min in a pre-warmed TLA-120 rotor (Beckman-Coulter). Supernatant was carefully removed, and the pellets were gently washed 4 times with pre-warmed assembly

buffer, and then resuspended with 6 M guanidine to denature the protein and release bound nucleotide. After 10-fold dilution into water, samples were loaded onto a Cellulose PEI TLC plate (Selecto Scientific) and TLC was performed, first with water followed by buffer containing 0.75 M Tris, 0.4 M LiCl, and 0.45 M HCl. The TLC plate was exposed to X-ray film after air drying. Radiolabelled mixtures of GTP/GDP and GTP/Pi were used as markers.

## Analytical ultracentrifugation

Samples for analytical ultracentrifugation (TOG1, TOG2, yeast αβ-tubulin, and mutant T238A polymerization-competent yeast αβ-tubulin) were dialyzed into final buffer conditions of RB100 (25 mM Tris pH 7.5, 1 mM $MgCl_2$, 1 mM EGTA, 100 mM NaCl) containing 20 μM GTP. The samples shown in *Figure 3* contain 0.15 μM T238A yeast αβ-tubulin and 0.015 μM, 0.03 μM, 0.15 μM, 0.3 μM, 0.6 μM, 1.2 μM, 5 μM TOG1 or 0.015 μM, 0.025 μM, 0.05 μM, 0.1 μM, 0.2 μM, 0.4 μM, 0.8 μM, 1.6 μM, and 5 μM TOG2. Samples were mixed and incubated at 4°C for at least one hr prior to the experiment. All analytical ultracentrifugation experiments were carried out in an Optima XL-I centrifuge using an An50-Ti rotor (Beckman-Coulter). Approximately 390 μL of each sample were placed in charcoal-filled, dual-sector Epon centerpieces. Sedimentation (rotor speed: 50,000 rpm) was monitored using absorbance at 229 nm and centrifugation was conducted at 20°C after the centrifugation rotor and cells had equilibrated at that temperature for at least 2.5 hrs. Protein partial-specific volumes, buffer viscosities, and buffer densities were calculated using SEDNTERP (*Laue et al., 1992*). Data were analyzed using SEDFIT and SEDPHAT (available at http://www.analyticalultracentrifugation.com) (*Schuck, 2000*; *Schuck et al., 2002*).

## Microtubule depolymerization fluorescence assays

Wild-type and mutant yeast αβ-tubulin were labeled with 6-–10 μM ReAsH in DMSO (Life Technologies) and 1 mM TCEP, in tubulin storage buffer (see above) for 90 min at RT. To remove excess unbound dye, samples were exchanged into assembly buffer (100 mM PIPES pH 6.9, 10% glycerol, 2 mM $MgSO_4$, 0.5 mM EGTA) with 2 mL, 7K MWCO Zeba spin desalting columns (Thermo Scientific). Labeled wild-type and mutant yeast αβ-tubulin were polymerized in the presence of 3–5 μM epothilone-B in assembly buffer (see above) with 1 mM GTP or in the presence of 500 μM GTPγS in assembly buffer (see above). The mixture was incubated for 30 min at 30 °C.

Flow chambers were prepared as described above. His-Tag Antibody (1:200, Gentech) was incubated in the chamber for 10 min, followed by incubation with 1% Pluronic F-127 in BRB80 for 5 min, followed by a wash with BRB80. Pre-formed, epothilone- or GTPγS-stabilized wild-type or T238A yeast MTs were then introduced into the chamber and allowed to incubate for 10 min, followed by a wash with BRB80 to remove unbound MTs. Solutions containing a range of TOG1 (0.1-–10 μM) or TOG1(R200A) (250, 750 nM) concentrations (*Figure 5B*) in imaging buffer (BRB80 + 200 nM epothilone + 0.1 mg/mL BSA + antifade reagents (glucose, glucose oxidase, catalase), without the addition of β-mercaptoethanol [*Gell et al., 2010*]) were introduced into the chamber immediately prior to data collection. To ensure that the TOG-induced depolymerization was not an artifact of fluorescence imaging, unlabeled wild-type microtubule were assembled as described above but without ReAsH labeling and then imaged with two concentrations of TOG1 using DIC microscopy (black points in *Figure 5B*). MT depolymerization reactions with fluorescent microtubules were imaged by epifluoresence microscopy using an Olympus IX81 microscope with a Plan Apo N 60x/1.42 NA objective lens and Hamamatsu ORCA-Flash2.8 CMOS camera, a mercury short arc lamp, and a Texas Red filter cube (Olympus). Reactions were temperature controlled and the microscope was controlled as described above. Images of MT shrinking were recorded every 60 s for about 10 min. MT depolymerization reactions with non-fluorescent wild-type microtubules were imaged using differential interference microscopy as described above. MT length was measured manually using ImageJ (*Schneider et al., 2012*). Average lengths of MT's were taken over the time course of the movie and were used to determine rate of depolymerization over an hour time span.

## Dynamic assays with Bim1-GFP

Preparation of flow chambers using sea urchin axonemes was followed as described above. Samples of wild-type, β:T238A, or C354A αβ-tubulin with 1 mM GTP or GTPγS were prepared and incubated in flow-chambers for 30–180 min and observed under DIC to evaluate presence of microtubule

growth. After a given time, solutions containing desired tubulin sample and nucleotide of interest, along with 5 or 50 nM Bim1-GFP with imaging buffer described above, were flowed into the chamber. Interactions of Bim1-GFP with MTs were imaged by total internal reflection fluorescence microscopy using an Olympus IX81 microscope with a TIRF ApoN 60x/1.49 objective lens, a 491 nm 50 mW solid-state laser and Hamamatsu ORCA-Flash2.8 CMOS camera (Olympus). Reactions were temperature controlled at 30°C and the microscope was controlled as described above. Axonemes were tracked under DIC and TIRF conditions. Images of MTs were taken over several frames from 15–30 min. Bim1-GFP fluorescence intensity along microtubules and extending beyond their growing ends was obtained using the PlotProfile function in ImageJ (*Schneider et al., 2012*). These linescans were manually aligned to superimpose the sharp change in intensity at the very microtubule end, and aligned intensity values were averaged.

### Analysis of sequence conservation

To collect a large number of α- and β-tubulin sequences, we performed psiblast (*Altschul et al., 1997*) (non-redundant (nr) database, 3 iterations) on the amino acid sequences of Tub1p and Tub2p from *S. cerevisiae*. The top 1000 sequences from both searches were aligned using Clustal Omega (*Sievers et al., 2011*) and manually pruned in Jalview (*Waterhouse et al., 2009*) to eliminate sequences with large insertions or deletions.

## Acknowledgements

We thank C. Brautigam in the UT Southwestern Macromolecular Biophysics Resource for help and advice. S Bechstedt in G Brouhard's lab performed pilot experiments that pointed us to the TOG-induced depolymerization assay. Electron microscopy was performed in the UT Southwestern Electron Microscopy Core Facility. H Yu, X Zhang, B Russ, G Brouhard, J Kollman, and C Asbury gave insightful comments on the manuscript. LMR is the Thomas O Hicks Scholar in Medical Research. FAP and EAG. were supported by NIH T32 GM008297, and EAG. was supported by an NSF Graduate Research Fellowship, Grant No. 2014177758. This material is based upon work supported by the National Science Foundation Graduate Research Fellowship under Grant No. 2014177758. Any opinion, findings, and conclusions or recommendations expressed in this material are those of the authors(s) and do not necessarily reflect the views of the National Science Foundation. A. B. was a UT Southwestern Medical Center/UT Dallas Green Fellow. This work was supported by GM-098543 (from the NIH), and MCB-1054947 (from the NSF).

## Additional information

### Funding

| Funder | Grant reference number | Author |
| --- | --- | --- |
| National Institute of General Medical Sciences | GM-098543 | Luke M Rice |
| National Science Foundation | MCB-1054947 | Luke M Rice |
| National Science Foundation | 2014177758 | Elisabeth A Geyer |
| National Institute of General Medical Sciences | T32 GM008297 | Elisabeth A Geyer Felipe-Andres Piedra |

The funders had no role in study design, data collection and interpretation, or the decision to submit the work for publication.

### Author contributions

EAG, TCH, LMR, Conception and design, Acquisition of data, Analysis and interpretation of data, Drafting or revising the article; AB, XY, Acquisition of data, Analysis and interpretation of data, Drafting or revising the article; BAL, Acquisition of data, Analysis and interpretation of data; FAP, Analysis and interpretation of data, Drafting or revising the article, Contributed unpublished essential data or reagents

## Additional files

### Major datasets

The following previously published datasets were used:

| Author(s) | Year | Dataset title | Dataset URL | Database, license, and accessibility information |
|---|---|---|---|---|
| Lowe J, Li H, Downing KH, Nogales E | 2001 | Refined structure of alpha-beta tubulin from zinc-induced sheets stabilized with taxol | http://www.rcsb.org/pdb/explore/explore.do?structureId=1JFF | Publicly available at the RCSB Protein Data Bank (Accession no: 1JFF). |
| Ayaz P, Ye X, Huddleston P, Brautigam CA, Rice LM | 2012 | A TOG:alpha/beta-tubulin complex structure reveals conformation-based mechanisms for a microtubule polymerase | http://www.rcsb.org/pdb/explore.do?structureId=4FFB | Publicly available at the RCSB Protein Data Bank (Accession no: 4FFB). |

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
