## [Decision Letter]

Thank you for submitting your work entitled "A mutation uncouples the tubulin conformational and GTPase cycles, revealing allosteric control of microtubule dynamics" for peer review at *eLife*. Your submission has been favorably evaluated by Vivek Malhotra (Senior editor), a guest Reviewing editor, and three reviewers.

The reviewers have discussed the reviews with one another and the Reviewing editor has drafted this decision to help you prepare a revised submission.

Summary:

This manuscript describes the characterization of a T238A mutation in beta-tubulin of *S. cerevisiae*. In mutant cells, the microtubules are highly stabilized, and the purified mutant tubulin forms microtubules that depolymerize very slowly. The authors also determine that subsequent to polymerization, GTP is however hydrolyzed by mutant tubulin more efficiently than in microtubules polymerized from wild type tubulin. Surprisingly, EB1 decoration is not restricted to the microtubule end, suggesting that the conformational change in response to GTP hydrolysis in this mutant is different from that in normal microtubules. Similar behavior is observed for another mutation in helix 7 of beta-tubulin, consistent with prior reports (e.g. Alushin et al, 2014) that this helix is involved in the conformational changes in tubulin following hydrolysis of GTP. This is the first thorough in vitro characterization of a mutation in tubulin that uncouples GTPase cycle reactions from a conformational transition.

Essential revisions:

Although the majority of reviewers found this study very interesting and significant, some concerns were raised regarding the mechanistic insight, the way the manuscript presents its major conclusions and some technical issues of the assays as presented. These issues need to be addressed and are listed below.

Specific points of major criticism:

I) Presentation of conclusions: In its generality the novelty of the observed uncoupling of GTPase transitions and allosteric conformational changes appeared somewhat exaggerated given that drugs like taxol are well known to cause such uncoupling; this should be acknowledged and the conclusions stated more precisely. At times observations were not clearly separated from interpretations (TOG binding–conformational changes); more precise language is encouraged throughout. At times relevant literature was not acknowledged: high GTP content in yeast tubulin: Dougherty et al, Biochemistry 37, 1998, p. 10861; importance of the H7 helix for microtubule stability: Amos, Org Biomol Chem 2, 2004, p. 2153.

II) Technical quality/support of conclusions by the data: Several technical concerns have been identified that require additional experiments:

1) Figure 3 – TOG binding curves. The quality of the assay was not considered sufficient (large errors, saturation not reached) to support the strong statement that the mutant does not change TOG binding, leading to the interpretation that it does not change the conformation of unpolymerised tubulin. Additional data points need to be collected to bring the errors down to support such a strong statement convincingly.

2) Figure 4 – Bim1 end tracking: More example of images or kymographs of Bim1 end tracking on wildtype and mutant microtubules should be shown to demonstrate how 'typical' behavior looks like under the conditions used. Bim1 binding to mutant microtubules is only shown for one Bim1 concentration. Therefore, it remains unclear if simply the overall affinity of Bim1 for microtubule binding is increased or if indeed the relative accumulation at microtubule ends compared to the lattice is reduced/is now absent (as the authors conclude). Experiments with different Bim1 concentrations are needed to distinguish between these two possibilities (effect on overall affinity versus end binding selectivity).

3) Figure 5 – depolymerization of mutant microtubules by TOG: Technical concerns regarding the assay conditions have been raised. It was considered possible that the strong depolymerization activity of TOG could maybe be a consequence of artefacts due to singlet oxygen generation by ReAsH in the absence of reducing agents/strong oxygen scavenging system. The possibility of such effects should be excluded experimentally (changing overall fluorescence excitation conditions, adding oxygen scavengers, or changing fluorophore).

4) Figure 1 – dependence of growth velocities on tubulin concentration: the data for mutant versus wildtype microtubules have been collected in very different tubulin concentration regimes (due to the strong nucleation of mutant tubulin). So the strong conclusion that this dependence is identical for both types of microtubules relies on how reliably the individual curves can be extrapolated into the other regime/how big the errors of the fits are. These errors should be examined in more detail. Ideally, an attempt should be made to collect data points for wildtype microtubules close to its critical concentration using a recently published method (Wieczorek et al., Nat Cell Biol 17, 2015, p. 907).

5) One reviewer felt strongly that simple negative stain EM should be performed to demonstrate that the microtubule structure appears normal for mutant microtubules. Abnormalities not visible by fluorescence microscopy can then be excluded. This would set the standard for other future studies with mutant tubulins.

[Editors' note: further revisions were requested prior to acceptance, as described below.]

Thank you for resubmitting your work entitled "A mutation uncouples the tubulin conformational and GTPase cycles, revealing allosteric control of microtubule dynamics" for further consideration at *eLife*. Your revised article has been favorably evaluated by Vivek Malhotra (Senior editor) and a guest Reviewing editor. The manuscript has been improved but there are some remaining issues that need to be addressed before acceptance, as outlined below.

Most major points of criticism have been addressed satisfactorily by additional experiments and the presentation has been improved. At several instances however the clarity of presentation would still profit from further improvement and one essential control experiment is missing:

I) Missing control experiment:

Figure 4: The new data clearly strengthen the manuscript. The schemes and the language used here continue to suggest that the GTPgS lattice is identical to the mutant lattice. However, the new data demonstrate that there is a difference between the mutant GDP microtubule body and the GTP end. This may well indicate that there may also be a difference between the mutant GDP body and the GTPgS body (GTPgS being potentially more similar to the end than to the mutant body). A control experiment with 5 nM Bim1 on GTPgS microtubules is missing. The result of this experiment might require the authors to modify the scheme and adopt a more differentiated view about the microtubule lattice conformations of wt, mutant and GTPgS microtubules.

II) Presentation:

1) Abstract and text: The language of the authors continues to imply complete uncoupling between GTPase reaction and conformational changes, although the new Bim1 data in Figure 4 demonstrate that there is clearly a conformational change in the mutant in response to GTPase, however at a reduced level as compared to wild type. It is desirable to adjust language to reflect this point. For the same reason, the claim made in the Abstract that "post-GTPase conformational changes are not strictly required for catastrophe" does not appear to be supported by the new data and it is recommended to adjust language.

2) Yeast strains: The genotypes of the strains used for the tubulin productions and live cell microscopy experiments should be stated (Methods, Legends and/or Table).

3) Figure 1: The number of observed events used for the quantifications does not appear to be stated. Please check throughout if all numbers relevant for statistical analysis are provided.

4) Figure 1, Figure 6: Please add information to the figures to orient the reader where the inside/outside, plus/minus end of the microtubule is.

5) Figure 1: Please provide clearer examples of kymographs. Contrast is very low.

6) Figure 2: The tubulin concentrations used for the data presented in Figure 2 does not seem to be provided. Please put this concentration into context of the concentrations used in A and the discussion about the oligomers that might lead to the measurement of an increased GTP level.

7) Figure 4: Please explain how alignment and averaging of the Bim1 intensity profiles was performed. Please clarify if error bars are shown (the GTPgS graph displays some fuzzy haze which might be bars) and adjust the display so that the data can be seen clearly. Please check the manuscript throughout and state which errors are shown (s.d. or s.e.m.).

8) Figure 5: "Similar results were obtained using MTs stabilized with GTPgS (not shown)." Please show the existing data to support the claim.

9) The conclusions drawn here should be discussed in light of a recent publication from the Nogales lab (Cell 162, 849, 2015) which contains relevant information about the structure of microtubules in the absence and presence of an end binding protein and different nucleotides. Some of the interpretations might need refinement.

---

## [Author Response]

*Although the majority of reviewers found this study very interesting and significant, some concerns were raised regarding the mechanistic insight, the way the manuscript presents its major conclusions and some technical issues of the assays as presented. These issues need to be addressed and are listed below.*

We appreciate that the reviewers found our work to be interesting and significant. In brief we have tuned and tightened the language throughout, and we have added new data and/or panels to Figure 1, Figure 3, Figure 4, and 5. The revised manuscript is stronger and clearer, and substantively addresses all of the concerns articulated in the decision letter.

*Specific points of major criticism: I) Presentation of conclusions: In its generality the novelty of the observed uncoupling of GTPase transitions and allosteric conformational changes appeared somewhat exaggerated given that drugs like taxol are well known to cause such uncoupling; this should be acknowledged and the conclusions stated more precisely. At times observations were not clearly separated from interpretations (TOG binding–conformational changes); more precise language is encouraged throughout. At times relevant literature was not acknowledged: high GTP content in yeast tubulin: Dougherty et al, Biochemistry 37, 1998, p. 10861; importance of the H7 helix for microtubule stability: Amos, Org Biomol Chem 2, 2004, p. 2153.*

We completely agree that small molecules such as taxol are already known to uncouple the GTPase and conformational cycles, and we did not intend to appear to lay claim to having discovered this for the first time. We have changed the Abstract, Introduction, Results, and Discussion sections to be clearer and more precise about what is different and novel in our work.

We have also added the suggested citations and also tightened the language throughout.

*II) Technical quality/support of conclusions by the data: Several technical concerns have been identified that require additional experiments: 1) Figure 3 – TOG binding curves. The quality of the assay was not considered sufficient (large errors, saturation not reached) to support the strong statement that the mutant does not change TOG binding, leading to the interpretation that it does not change the conformation of unpolymerised tubulin. Additional data points need to be collected to bring the errors down to support such a strong statement convincingly.*

We have collected additional data points for the TOG2 titration, which slightly increased the fitted affinity and slightly decreased its error range. Lack of saturation was not a factor. A simple thought experiment will illuminate why these data are sufficient to support an argument that the mutation does not lead to substantial change in αβ−tubulin curvature. Based on our measurements, TOG domains bind approximately 2-fold tighter to β:T238A αβ-tubulin than they do to wild-type. A 2-fold difference in binding constant is equivalent to about 0.7 kT of binding energy – this is a similar order of magnitude as one hydrogen bond, and therefore hard to square with a substantial conformational rearrangement in the mutant. We can also estimate the order of magnitude change in binding affinity we might expect for curved vs straight αβ-tubulin. ~100 nM affinity corresponds to about 16 kT of total binding energy. TOG domains contact both α- and β- tubulin when binding to curved αβ-tubulin, but in straight αβ-tubulin TOG domains cannot maintain both contacts simultaneously. Making a simplifying assumption that about half of the binding energy (8 kT) will be lost for the straight conformation would correspond to ~3000-fold weaker binding, an estimate that is consistent with the fact that we have not observed significant binding of TOGs to microtubules or detectable binding to αβ-tubulin by a ‘half-site’ TOG1 mutant (R200A). Thus, the fitted affinities and their associated errors are not consistent with the idea that there is a substantial mutation-induced change in TOG binding affinity, and this in turn supports our claim that the conformation of the mutant cannot be very different than that of the wild-type. We feel that this is especially so given that we readily detected a change in Bim1 binding affinity for the body of mutant microtubules, and that we also observed a nearly two order of magnitude change in depolymerization rate, all with only small detectable changes in the apparent biochemistry of elongation.

In the original submission, we had tried to be careful throughout the manuscript to state that the mutant did not detectably change the conformation of unpolymerized αβ-tubulin. In the revised manuscript we have maintained the use of qualifiers to indicate the lack of complete certainty (which would likely require structures of mutant and wild-type αβ- tubulin, and more) while also re-working the discussion surrounding these data to include some of the arguments from the prior paragraph. We appreciate the chance to clarify these issues.

*2) Figure 4 – Bim1 end tracking: More example of images or kymographs of Bim1 end tracking on wildtype and mutant microtubules should be shown to demonstrate how 'typical' behavior looks like under the conditions used.*

We have now included more images of Bim1 end tracking on wild-type and mutant microtubules (4 for each condition as opposed to 1 previously; we also added two new conditions, see below).

*Bim1 binding to mutant microtubules is only shown for one Bim1 concentration. Therefore, it remains unclear if simply the overall affinity of Bim1 for microtubule binding is increased or if indeed the relative accumulation at microtubule ends compared to the lattice is reduced/is now absent (as the authors conclude). Experiments with different Bim1 concentrations are needed to distinguish between these two possibilities (effect on overall affinity versus end binding selectivity).*

This comment raises interesting ideas about the microtubule cap. We have performed additional experiments in response to it. In our initial submission the Bim1 binding clearly showed that the body of mutant microtubules presents higher-affinity Bim1 binding sites than does the body of wild-type microtubules. This comment asks us to determine if the same is true in the cap region. New experiments using 5 nM Bim1 (10-fold lower concentration than in the original submission) show that (i) the Bim1 intensity in the cap is comparable for mutant and wild-type microtubules, suggesting that wild-type and mutant caps present Bim1 binding sites of comparable affinity, (ii) the Bim1 intensity on the body of mutant microtubules is significantly higher than on the body of wild-type microtubules, consistent with our proposal that αβ-tubulin adopts a more expanded conformation in β:T238A microtubules, and (iii) we see ‘comets’ on the end of mutant microtubules, indicating that Bim1 binds more tightly to the ends of mutant microtubules than to the body.

Our bottom-line conclusion that β:T238A αβ-tubulin adopts a more expanded, GTP-like conformation in a GDP lattice remains unchanged. However, by showing that Bim1 binds to mutant microtubule caps more tightly than to mutant microtubule bodies, the new data now make it very clear that the mutation does not completely override the nucleotide effect. We have modified the Discussion to reflect these new findings, including substituting ‘suppress’ for more absolute words like ‘eliminate’. In our view these new data and associated discussion strengthen the manuscript, and we are grateful for the push to do these experiments.

*3) Figure 5 – depolymerization of mutant microtubules by TOG: Technical concerns regarding the assay conditions have been raised. It was considered possible that the strong depolymerization activity of TOG could maybe be a consequence of artefacts due to singlet oxygen generation by ReAsH in the absence of reducing agents/strong oxygen scavenging system. The possibility of such effects should be excluded experimentally (changing overall fluorescence excitation conditions, adding oxygen scavengers, or changing fluorophore).*

The assays reported in our initial submission did in fact include a strong oxygen scavenging system (Glucose oxidase + catalase), but we can see how this might have been overlooked because we lumped them into the phrase ‘antifade reagents’. We now mention these reagents explicitly, and we also performed two kinds of control experiments to rule out the possibility of a non-specific origin of the effects we observed. First, we monitored the rate of TOG-induced shrinking using differential interference contrast microscopy at two different TOG concentrations and without any fluorescent labels. These experiments gave microtubule shrinking rates consistent with what we initially reported. Second, using the ReAsH labeled microtubules, we performed experiments using a TOG domain containing a mutation that greatly reduces tubulinbinding affinity. These experiments revealed greatly reduced rates of TOG-induced shrinking. These new data have been added to Figure 5. Together, they rule out the possibility that the TOG-induced depolymerization was an artifact.

*4) Figure 1 – dependence of growth velocities on tubulin concentration: the data for mutant versus wildtype microtubules have been collected in very different tubulin concentration regimes (due to the strong nucleation of mutant tubulin). So the strong conclusion that this dependence is identical for both types of microtubules relies on how reliably the individual curves can be extrapolated into the other regime/how big the errors of the fits are. These errors should be examined in more detail. Ideally, an attempt should be made to collect data points for wildtype microtubules close to its critical concentration using a recently published method (Wieczorek et al., Nat Cell Biol 17, 2015, p. 907).*

We had examined the fitting errors but somehow managed to not include this in the manuscript. We have now corrected this oversight by including the fitting errors and by discussing them when we are comparing the slopes and intercepts of the two fitted lines. To minimize the ‘distance’ of extrapolation, we also obtained an additional data point for wild-type αβ-tubulin using the Wieczorek/Brouhard method. We apologize for not including and discussing fitting errors in our initial submission.

5) One reviewer felt strongly that simple negative stain EM should be performed to demonstrate that the microtubule structure appears normal for mutant microtubules. Abnormalities not visible by fluorescence microscopy can then be excluded. This would set the standard for other future studies with mutant tubulins.

We have now included negative stain EM images showing that wild-type and β:T238A microtubules display similar structure.

[Editors' note: further revisions were requested prior to acceptance, as described below.]

*I) Missing control experiment: Figure 4: The new data clearly strengthen the manuscript. The schemes and the language used here continue to suggest that the GTPgS lattice is identical to the mutant lattice. However, the new data demonstrate that there is a difference between the mutant GDP microtubule body and the GTP end. This may well indicate that there may also be a difference between the mutant GDP body and the GTPgS body (GTPgS being potentially more similar to the end than to the mutant body). A control experiment with 5 nM Bim1 on GTPgS microtubules is missing. The result of this experiment might require the authors to modify the scheme and adopt a more differentiated view about the microtubule lattice conformations of wt, mutant and GTPgS microtubules.*

We had chosen to focus on the differential conformational response to GDP, so we did not perceive the 5 nM Bim1 on GTPgS microtubules as a missing control because GTPgS microtubules have uniform nucleotide state. We now show those data in the revised Figure 4, and they document the expected result that Bim1 coats the body of GTPgS microtubules with similar intensity as it does the tip. To avoid conveying an impression that the mutant lattice is identical to a GTP/GTPgS lattice, we have made changes to the language throughout the paper (see also our response to the next point).

*II) Presentation: 1) Abstract and text: The language of the authors continues to imply complete uncoupling between GTPase reaction and conformational changes, although the new Bim1 data in Figure 4 demonstrate that there is clearly a conformational change in the mutant in response to GTPase, however at a reduced level as compared to wild type. It is desirable to adjust language to reflect this point. For the same reason, the claim made in the Abstract that "post-GTPase conformational changes are not strictly required for catastrophe" does not appear to be supported by the new data and it is recommended to adjust language.*

In the revised text we submitted we had inserted a number of qualifying statements in an effort to not portray the ‘uncoupling’ in a black and white manner. We have now done even more of that, adopting phrasing that is closer to the biochemical finding (e.g. stronger Bim1 binding, GTP-lattice-like character) in order to avoid too many statements about specific conformations (expanded/compacted). We also modified the ‘strictly required’ sentence in the Abstract, which we should have done the first time around. Thanks for catching that, and apologies for missing it.

*2) Yeast strains: The genotypes of the strains used for the tubulin productions and live cell microscopy experiments should be stated (Methods, Legends and/or Table).*

Strain genotypes have been added as a separate section in the Methods.

*3) Figure 1: The number of observed events used for the quantifications does not appear to be stated. Please check throughout if all numbers relevant for statistical analysis are provided.*

Done in Figure 1 and throughout. As part of this checking we changed some panels in Figure 1 to show s.e.m instead of s.d.

*4) Figure 1, Figure 6: Please add information to the figures to orient the reader where the inside/outside, plus/minus end of the microtubule is.*

We have explained the viewing direction in the legends of the figures where we show views of the structures.

*5) Figure 1: Please provide clearer examples of kymographs. Contrast is very low.*

Done.

*6) Figure 2: The tubulin concentrations used for the data presented in Figure 2 does not seem to be provided. Please put this concentration into context of the concentrations used in A and the discussion about the oligomers that might lead to the measurement of an increased GTP level.*

We have listed the concentrations in the Methods and in the figure legend and in the text, where we also note that they are within the range shown in panel A. We did not add additional discussion, in part because panels A, B, and C each used different centrifugation settings. We did adjust the language we used to discuss possible origins of the increased GTP level.

*7) Figure 4: Please explain how alignment and averaging of the Bim1 intensity profiles was performed. Please clarify if error bars are shown (the GTPgS graph displays some fuzzy haze which might be bars) and adjust the display so that the data can be seen clearly. Please check the manuscript throughout and state which errors are shown (s.d. or s.e.m.).*

We have added to the methods section to better explain what we did. We agree that the error bars were not clearly visible and we have altered the formatting to improve the presentation of the data. Legends now state if error bars represent s.d or s.e.m.

*8) Figure 5: "Similar results were obtained using MTs stabilized with GTPgS (not shown)." Please show the existing data to support the claim.*

We have added these data points to Figure 5.

*9) The conclusions drawn here should be discussed in light of a recent publication from the Nogales lab (Cell 162, 849, 2015) which contains relevant information about the structure of microtubules in the absence and presence of an end binding protein and different nucleotides. Some of the interpretations might need refinement.*

Two factors make it difficult to draw direct comparisons between our results and the three lattice structures recently described by Eva Nogales and colleagues. First, we don’t yet have high-resolution structures for wild-type or mutant yeast microtubules, so we can’t know exactly which conformations are involved. Second, yeast microtubules assemble readily using GTPgS but less so with GMPCPP, whereas the opposite is true for vertebrate microtubules – GTPgS may therefore be promoting different states for the two kinds of microtubules. These factors limit our ability to make detailed structural interpretations, and as mentioned above we have made changes to our language to try to make this clear. Nevertheless, the different Bim1 binding properties of wild-type and mutant microtubules provide strong support for the idea that the mutant retains significant GTP-lattice-like character in a GDP lattice. Encouraging more explanation around these ideas was a good suggestion, and we have added a few sentences along the lines of the above to the end of the Discussion.